# Increased Sulfation in *Gracilaria fisheri* Sulfated Galactans Enhances Antioxidant and Antiurolithiatic Activities and Protects HK-2 Cell Death Induced by Sodium Oxalate

**DOI:** 10.3390/md20060382

**Published:** 2022-06-07

**Authors:** Waraporn Sakaew, Jenjiralai Phanphak, Somsuda Somintara, Wiphawi Hipkaeo, Kanokpan Wongprasert, José Kovensky, Choowadee Pariwatthanakun, Tawut Rudtanatip

**Affiliations:** 1Electron Microscopy Unit, Department of Anatomy, Faculty of Medicine, Khon Kaen University, Khon Kaen 40002, Thailand; warapsa@kku.ac.th (W.S.); jenjilalai.phanphakdee@gmail.com (J.P.); tsomsuda@kku.ac.th (S.S.); wiphawi@kku.ac.th (W.H.); 2Department of Anatomy, Faculty of Science, Mahidol University, Bangkok 10400, Thailand; kanokpan.won@mahidol.ac.th; 3Laboratoire de Glycochimie, des Antimicrobiens et des Agroressources (LG2A) CNRS UMR 7378, Institut de Chimie de Picardie FR 3085, Université de Picardie Jules Verne, 80000 Amiens, France; jose.kovensky@u-picardie.fr; 4Division of Anatomy, Faculty of Medicine, Mahasarakham University, Mahasarakham 44000, Thailand; choowadee@msu.ac.th

**Keywords:** sulfated galactans, sulfation, antioxidant, antiurolithiasis, sodium oxalate

## Abstract

Urolithiasis is a common urological disease characterized by the presence of a stone anywhere along the urinary tract. The major component of such stones is calcium oxalate, and reactive oxygen species act as an essential mediator of calcium oxalate crystallization. Previous studies have demonstrated the antioxidant and antiurolithiatic activities of sulfated polysaccharides. In this study, native sulfated galactans (N-SGs) with a molecular weight of 217.4 kDa from *Gracilaria fisheri* were modified to obtain lower molecular weight SG (L-SG) and also subjected to sulfation SG (S-SG). The in vitro antioxidant and antiurolithiatic activities of the modified substances and their ability to protect against sodium oxalate-induced renal tubular (HK-2) cell death were investigated. The results revealed that S-SG showed more pronounced antioxidant activities (DPPH and O_2_^−^ scavenging activities) than those of other compounds. S-SG exhibited the highest antiurolithiatic activity in terms of nucleation and aggregation, as well as crystal morphology and size. Moreover, S-SG showed improved cell survival and increased anti-apoptotic BCL-2 protein in HK-2 cells treated with sodium oxalate. Our findings highlight the potential application of S-SG in the functional food and pharmaceutical industries.

## 1. Introduction

Urolithiasis is a frequently occurring urological disease characterized by the presence of a stone anywhere along the urinary tract, with a high recurrence rate [1]. Arunkajohnsak et al. found that the major components of the stone consist of calcium oxalate (60%), calcium phosphate (26%), uric acid (8.3%), magnesium ammonium phosphate (4.0%), cystine (1.0%), and ammonium hydrogen urate (0.6%) [2]. The pathogenesis is a complex biochemical process that is not completely understood. It is generally accepted that stone formation depends on the level of imbalance between urinary inhibitors and the promoters of crystallization. The crystallization of the stone begins with the supersaturation of urine, which is excreted in stone formers more than in non-stone formers and is generally higher in patients with recurrent kidney stones than in those without kidney stones [3]. Supersaturation occurs when urine contains more dissolved material, such as calcium oxalate and phosphate, resulting in the formation of stone crystals. The transformation from the liquid to solid phase in urinary supersaturation is called nucleation. This is followed by crystal growth, in which the loose cluster may increase in size by adding new crystal components and aggregation, in which the crystals adhered to one another to form a larger mass. They then attached to renal tubular epithelial cells [4]. Injecting animals with sodium oxalate induces tubular calcium oxalate crystal excretion, followed by a decline in kidney function [5]. The high level of oxalate can cause an increase in the production of reactive oxygen species (ROS). These ROS also increase the amount of oxalate crystals, which enhance the formation of nucleation, crystal growth, and aggregation. Furthermore, it has been demonstrated that ROS-related oxalates induced renal tubular cell injury and death [6]. A previous study confirmed that antioxidants can reduce the size of calcium oxalate crystals and prevent them from being deposited, as well as induce a decrease in renal membrane damage induced by oxalate [7]. It is, thus, important to find compounds that have antioxidant activities and inhibit stone formation. In Eastern medicine, seaweed has been widely used to treat and/or prevent the damage caused by the calcium oxalate crystal formation in the urinary tract [8]. Polysaccharides, a major component of seaweed, have been shown to have antiurolithiatic activities. They inhibit the nucleation and aggregation of calcium oxalate crystallization and modify the morphology, size, and surface charge of calcium oxalate crystals [9]. Moreover, it has been reported that sulfated polysaccharides from brown seaweed *Fucus vesiculosus* decreased oxalate toxicity on renal cells through its antioxidant properties [10].

The structures of polysaccharides affect their biological activities, particularly those associated with molecular weight, monosaccharide composition, and other functional groups [11,12]. For instance, microalgal *Pavlova viridis* polysaccharides cannot exert biological activities because of their large molecular size and poor solubility. In addition, *P. viridis* polysaccharides exhibit higher immunomodulation and antitumor activities after degradation, which are correlated with low molecular weight after modification [13]. The low molecular weight mushroom *Tremella fuciformis* polysaccharide exhibits greater scavenging activity than its natural counterpart [14]. Conceptually, the degradation of high into low molecular weight polysaccharides might allow them to more easily soluble and exhibit better bioactivity [15].

Accumulated data have demonstrated that sulfated polysaccharides are highly related with bioactivity and that removing the sulfate contents leads to decreased bioactivity [15]. Sulfation is the process of adding sulfate groups to the hydroxyl groups of the polysaccharides, which has been shown to enhance their antioxidant and hypoglycemic activities [16]. Thus, the modification of the molecular structure of polysaccharides is of interest, as it could enhance the bioactivities of polysaccharides. Previous studies reported that sulfated galactans (SG) isolated from the marine red seaweed *Gracilaria fisheri* exhibited radical scavenging ability [17,18]. However, it is possible that antiurolithiatic abilities have not yet been investigated. Thus, we modified the structures of *G. fisheri* SG (N-SG) to lower their molecular weight (L-SG) and raised their sulfate content (S-SG). In addition, we assessed their antioxidant and antiurolithiatic activities, which inhibited calcium oxalate crystal formation and their ability to repair HK-2 cells with sodium oxalate-induced damage.

## 2. Results

### 2.1. Sulfate Content and Average Molecular Weight of N-SG and Its Derivatives

The sulfate contents of N-SG, L-SG, and S-SG were 11.37 ± 0.96%, 10.59 ± 0.06%, and 17.11 ± 0.21%, respectively. That of S-SG increased significantly after sulfated modification, indicating that the addition of the sulfate groups was successful. Moreover, the initial molecular weight was 217.45 kDa (N-SG) compared to 7.87 kDa after HCl degradation (L-SG). After sulfated modification, the molecular weight of S-SG was 323.83 kDa, confirming that the increase in size was due to sulfate increases in S-SG structure.

### 2.2. Antioxidant Activities of N-SG and Its Derivatives

Figure 1 shows the free-radical scavenging capacity of the compounds compared to ascorbic acid as a reference. The DPPH radical scavenging activity is shown in Figure 1a. The solution treated with S-SG had the highest DPPH scavenging activity at a concentration of 47.87 mg/g ascorbic acid equivalents compared to 7.9 mg/g ascorbic acid equivalents for N-SG and 2.64 mg/g ascorbic acid equivalents for L-SG. The superoxide (O_2_^−^) radical scavenging activity is shown in Figure 1b. O_2_^−^ scavenging activity was the highest in the solution treated with S-SG at a concentration of 111.89 mg/g ascorbic acid equivalents. On the other hand, those of N-SG and L-SG were observed at concentrations of 100.68 and 101.73 mg/g ascorbic acid equivalents, respectively.

### 2.3. Inhibitory Effect of N-SG and Its Derivatives on Calcium Oxalate Crystallization

The nucleation and aggregation of calcium oxalate crystals in solution were inhibited by the addition of N-SG, L-SG, and S-SG. We assessed the percentage of calcium oxalate crystal formation inhibition by nucleation and aggregation. The inhibition of calcium oxalate nucleation exhibited by the compounds and the standard drug Cystone was significantly (*p* < 0.05) higher than that of the control solution. The nucleation inhibition percentages of N-SG, L-SG, S-SG, and cystone were 39.57%, 48.00%, 97.38%, and 79.93%, respectively (Figure 2a). The order of potency for the compounds was S-SG > L-SG > N-SG. Interestingly, S-SG inhibited nucleation to a greater extent than drug Cystone.

Aggregation is also an important step that contributes to calcium oxalate crystal formation, which in turn aids in the progression and worsening of urolithiasis. An incubation of calcium oxalate crystals with the compounds resulted in reduced crystal aggregation. The compounds, as well as Cystone, inhibited calcium oxalate aggregation to a significantly (*p* < 0.05) greater degree than the control solution. S-SG had the highest aggregation inhibition percentage among the compounds at 27.26%, whereas those of N-SG, L-SG, and Cystone were 16.34%, 9.53%, and 35.17%, respectively. The order of activity for the compounds was S-SG > N-SG > L-SG (Figure 2b). S-SG was found to be the most active in terms of its antiurolithiatic effects. It exhibited the greatest inhibition of both nucleation and aggregation of calcium oxalate crystal formations.

### 2.4. Morphological Changes of Crystals after Treatment of N-SG and Its Derivatives

The formation of the calcium oxalate crystals in the presence of *G. fisheri* N-SG, L-SG, and S-SG, as well as of Cystone as the standard drug, was evaluated by the measurement of the crystal’s diameter. We found that treatments with N-SG, L-SG, and S-SG decreased crystal size when compared with the untreated control (Table 1 and Figure 3a), and this was significantly observed in N-SG, S-SG, and Cystone (*p* < 0.05). The mean diameter of the control crystals was 84.35 ± 3.16 nm, whereas those of N-SG, L-SG, S-SG, and cystone were 68.58 ± 6.70, 79.93 ± 3.02, 65.59 ± 6.71, and 26.50 ± 5.00 nm, respectively (Table 1).

Calcium oxalate crystals can be monohydrate (COM), dihydrate (COD), or trihydrate (COT), with the first being the main type present in urolithiasis. COT crystals are thermodynamically unstable and rarely observed. In this study, COM was the main type of crystal formation in synthetic urine. We found that N-SG and its derivatives were associated with the formation of COD crystals, as was Cystone (Table 1). COM crystals were deposited concentrically with radial striation, while COD crystals had a rectangular morphology (Figure 3b). N-SG, L-SG, S-SG, and cystone all induced the formation of both COM and COD crystals. The percentages of COD crystals in solutions treated with N-SG, L-SG, S-SG, and Cystone were 3.23%, 5.00%, 4.50%, and 12.08%, respectively, which were significantly (*p* < 0.05) higher than in the control solution (0.90%).

After the nucleation assay, LM and SEM were used identify the calcium oxalate crystals, as shown in Figure 3. As COM crystals were the main type found in the solution, we measured their size and observed their morphology after the nucleation phase was complete. In the control, COM crystals consisted of a conglomerate of multiple facets that elongated and radiated out from a central core. We found that N-SG, L-SG, and S-SG affected their size and morphology, which varied to a greater degree than in the control crystals. In the S-SG and N-SG treated groups, the facets that radiated out from the central core were shorter, and the edges of their faces were more rounded. In addition, the edges of crystals in the L-SG treated group were slightly rounded with blunt tips. The Cystone control crystals were the smallest. This reduction in facet length and the rounded angles of edges and tips may decreased interactions with renal epithelium. Moreover, the smaller crystals would pass more easily through the urinary tract.

### 2.5. Cytotoxicity of Sodium Oxalate (NaOX) and S-SG on HK-2 Cells

To assess the cytotoxicity of NaOX and S-SG, HK-2 cells were treated with different concentrations of 0.156 to 5.0 mmol/l for NaOX and 1 to 2000 μg/mL for S-SG. The results showed that NaOX exhibited a reduction in HK-2 cell proliferation in a dose-dependent manner. A significant reduction was observed beginning at 0.156 mmol and at 1.25 mmol, and cell viability showed less than 50% (Figure 4a). On the other hand, S-SG was non-cytotoxic at concentrations of 1–1000 μg/mL and significantly cytotoxic at more than 1500 μg/mL (Figure 4b).

### 2.6. Protective Effect of S-SG against NaOX-Induced HK-2 Cell Damage

To investigate whether S-SG protects against HK-2 cell damage induced by NaOX, we examined its effects of 1.25 mmol NaOX and S-SG at various concentrations on HK-2 cells at 24 h. The MTT results showed that S-SG at concentrations of 100 and 1000 μg/mL significantly (*p* < 0.05) protected HK-2 cell viability after stimulation with NaOX, while at 1 and 10 μg/mL, it had no protective effects (Figure 5a).

The morphological changes of the HK-2 treated cells were observed under phase contrast light microscope. As shown in Figure 5b, NaOX control cells showed less cell density and irregular shaped compared to control. In the cells treated with NaOX and followed with S-SG, cell density increased, and the degree of NaOX-induced cell damage decreased relative to S-SG concentrations. In terms of morphology, cells were plump and tightly connected in the 100-S-SG+NaOX and 1000-S-SG+NaOX groups, similarly to the control group.

### 2.7. Expression of Apoptosis-Related Proteins

Apoptosis is proposed by NaOX-induced HK-2 cell damage. The expression levels of apoptotic caspase-3 and anti-apoptotic BCL-2 proteins were determined by Western blotting (Figure 6). The results showed that the NaOX control group had significantly higher expressions of caspase-3 protein than those in the 100-S-SG+NaOX, 1000-S-SG+NaOX, and Cystone+NaOX groups. By contrast, treated cells in the 100-S-SG+NaOX, 1000-S-SG+NaOX, and Cystone+NaOX groups exhibited significantly higher BCL-2 protein expression compared with untreated control cells. This indicates that S-SG treatment modulated the expression of apoptosis-related proteins.

### 2.8. Structural Characterization of S-SG

The functional groups and chemical structures of N-SG and S-SG were confirmed by FTIR and ^1^H NMR analysis. The results of FTIR are shown in Figure 7a. At least eight prominent absorption peaks reflected the polysaccharides were observed. Peaks were present around 3364.09 and 2927.94 cm^−1^, which indicated the presence of OH- and CH-stretching vibrations. The peaks around 1639.49 and 1411.89 cm^−1^ were caused by the COO-antisymmetric stretching and the C=O symmetric stretching vibrations of the carboxylate group. The peak at approximately 1371.39 cm^−1^ was assigned to the sulfate ester bond. Moreover, the presence of sulfate esters was observed with peaks around 1249.87, 873.04, 856.39, 829.90, and 769.60 cm^−1^. With the comparation for N-SG, the FTIR spectrum of S-SG increased in the peaks at 873.04, 856.39, and 829.90 cm^−1^ assigned to galactose-6 sulfate, galactose-4 sulfate, and galactose-2 sulfate, respectively [19,20].

The ^1^H NMR spectra of N-SG and S-SG are shown in Figure 7b. Twelve chemical shift signals corresponding to β-D-galactose units (4.58, 3.67, 3.81, 4.14, 3.78, and 3.87 ppm) and 3,6-anhydrogalactopyranose (5.17, 4.15, 4.53, 4.68, 4.59, and 4.20 ppm) units of polysaccharides were observed [18]. After sulfation modification, the chemical shift signals of S-SG were enhanced and are different from those of N-SG. The chemical shift signal attributed to 6-sulfate on 4-linked α-L-galactopyranose was enhanced at 5.31 ppm. The methylated agarose on *O*-2 of 4-linked 3,6-anhydro-l-galactopylanose with sulfation on C-6 of D-galactopyranose was observed at 4.37 ppm [21]. The strong chemical shift signal appearing at 3.96 ppm indicated that the S-SG contained L-galactose-6-sulfate with sulfation on C-6 of D-galactopyranose [17,22]. Moreover, the chemical shift signals at 3.82 and 3.67 ppm attributed to methylation on *O*-6 of the 3-linked β-d-galactopyranose and on *O*-6 of the 3-linked β-d-galactopyranose with sulfation on C-6 of D-galactopyranose were also detected. Together, FTIR and NMR results indicated that sulfate ester was successfully added at the backbone structure of S-SG.

## 3. Discussion

Nucleation is the first step of stone formation, followed by growth and further aggregation, which can result in an occlusion of the urinary tract [4]. It has been demonstrated that increased reactive oxygen species (ROS) caused stone formation. Some studies demonstrated that treatment with potential antioxidants prevented stone formation [6]. In this study, sulfation sulfated galactans (S-SG) from the red seaweed *Gracilaria fisheri* exhibited strong free radical scavenging (DPPH and O_2_^−^) activity related to sulfate contents [18], which can inhibit the nucleation and aggregation of calcium oxalate crystals and modify their morphology. Many studies have indicated that sulfation is an effective method for enhancing the antioxidant activities of polysaccharides isolated from oyster *Crassostrea gigas* [23] and brown seaweed *Sargassum pallidum* [16]. There have also been reports describing the relationship between the sulfate content of polysaccharides and their antioxidant ability. Zhang et al. found that polysaccharides with high sulfate content from the brown seaweed *Porphyra haitanesis* showed high levels of superoxide radical scavenging activity [24]. The sulfate modification of polysaccharides extracted from mushroom *Pleurotus eous* enhanced DPPH radical scavenging activity by promoting the capture of electrons by DPPH. Because DPPH is an aromatic free radical that can bind to electrons or hydrogen ions, molecules with higher sulfate content can enhance the antioxidant capacity of polysaccharides [25]. In polysaccharides with similar molecular weight, higher sulfate content leads to stronger DPPH radical scavenging capacity at the same concentration. Previous studies indicate that the sulfate group is the most critical factor affecting the antioxidant activity of polysaccharides [26].

The nucleation of oxalate crystals is reportedly inhibited by polyanions such as proteins and polysaccharides. The negative charges of these molecules interact with Ca^2+^ ions to form soluble complexes, inhibit the formation of calcium oxalate crystals, and decrease the supersaturation of urine [27]. Polysaccharides such as S-SG may interact with the Ca^2+^ face of the crystals and act as an effective inhibitor of nucleation [28]. The interaction of calcium oxalate crystals and sulfated polysaccharide of the green seaweed *Caulerpa cupressoides* results in crystal surfaces exhibiting a more negative charge, which can decrease the size and the number of formed crystals [9]. On the other hand, the anionic molecules inhibit the formation of nuclei that prevented the aggregation phase. However, galactan from red seaweed *G. birdiae* increases the nucleation of calcium oxalate but inhibits aggregation by preventing the formation of large crystals [27]. In our study, the nucleation and aggregation of calcium oxalate crystals were most inhibited by the high sulfate content of S-SG from red seaweed *G. fisheri*. Moreover, S-SG changed the morphology and decreased the size of the crystals. In addition, L-SG was less effective than both N-SG and S-SG, although it has been reported that low molecular weight polysaccharides exhibit better absorption and bioavailability [29]. However, the biological activity of these polysaccharides could be reduced due to their being less likely to form a biologically active polymeric structure [30].

Calcium oxalate is the predominant crystalline component present in kidney stones. Calcium oxalate monohydrate (COM) is the most thermodynamically stable hydrate, followed by calcium oxalate dihydrate (COD), and calcium oxalate trihydrate (COT) is the most unstable [31]. COD crystals are less aggressive and are commonly present in the urine of people with asymptomatic urolithiasis. This urine contains molecules that prevent the formation of COM and stabilize COD [32]. Regarding crystal morphology, COM crystals have edges and tips with sharp angles. These morphological features are important factors that affect the binding of the COM crystals with renal tubular epithelium. It is important to emphasize that COM crystals attached with renal tubular epithelial cells result in crystal growth. The formation of large crystals on the renal tubular epithelial cell surface leads to the formation of kidney stones. On the other hand, COD crystals have less contact with the renal tubular epithelial cell surface [33,34]. Seaweed exhibits the ability to induce the formation of COD crystals or to inhibit the formation of COM crystals. For example, sulfated polysaccharides from marine seaweed *Dictyopteris justii* can inhibit the crystallization of calcium oxalate by stabilizing crystals as COD. Sulfated glucan is also able to prevent COD from turning into COM crystals [35]. Both native and modified sulfated polysaccharides extracted from Japanese honeysuckle *Lonicera japonica* have been shown to induce and stabilize the formation of COD crystals. Interestingly, modified sulfated polysaccharides has been shown to be more effective at inhibition than native sulfated polysaccharides [36]. The capacity of polysaccharides to repair damaged cells is highly positively correlated with sulfate content [37]. Previous studies have demonstrated that sulfation increases the water solubility of polysaccharides and significantly enhances their antioxidant abilities [38,39]. Moreover, cell–calcium oxalate crystal interaction is reduced due to the negative charge of the sulfate group covering the calcium oxalate surface [40]. The increased number of attached sulfate group from sulfation could improve the cell viability of damaged cells.

Upon interaction with oxalate crystals, cells generate ROS, which cause injury to the contacted cells [41,42]. Excessive ROS are produced when oxidative stress occurs in the kidneys, which leads to the apoptosis of the renal tubular epithelial cells [43,44]. Alterations of caspase-3 and BCL-2 proteins in HK-2 with NaOX-induced cell damage of S-SG indicated the protection of compounds from cell apoptosis. It has been reported that sodium oxalate causes mitochondrial dysfunction in renal cells [45] and mitochondrial damage and redox imbalance in THP-1 cells [42]. The cellular toxicity of oxalate on HK-2 cells occurs through apoptosis, as suggested by the increased expression of Bax and caspase 9 and decreased expression of BCL-2 protein [46]. Interestingly, stone flower *Didymocarpus pedicellata* reduces oxidative stress injury and apoptosis in HK-2 cells by reducing the production of ROS [47]. Polysaccharides alter oxalate crystal properties by covering the crystal surface [40,48] and altering the interaction between crystals and the cell receptor, thereby preventing crystal adhesion to the cells and reducing cell damage caused by the crystals [37]. Since crystals with rounded edges and tips have less surface area, they exhibit less adhesion to renal epithelial cells and are easily expelled out of the body along with urine [31].

Additionally, to confirm superior activity related with successful sulfate group supplementation, S-SG was chemically characterized. An increased percentage of sulfate groups after sulfation correlated with FTIR and NMR results. Sulfated galactans consist of the 3,6-anhydro bridge on the 4-linked galactose residue [17,49]. The position of sulfates is one of the main properties that determine the structure and function of sulfated polysaccharides. Previous studies have found that the sulfate groups of polysaccharides of edible red seaweeds (*Mastocarpus stellatus, Gigartina pistillata, Cymbovula acicularis, Nemalion helminthoides*, and *Dumontia contorta*) are represented by FTIR spectroscopic peaks at 750–1260 cm^−1^ [20], similarly to the absorption region found in S-SG in our study. In addition, the stretching vibration of sulfate ester bond in sulfated polysaccharides from *G. fisheri* is represented as a strong absorption peak at around 1360–1370 cm^−1^ [18]. ^1^H NMR spectroscopy indicated that a carbohydrate moiety of sulfated polysaccharides presents as intense signals of hydrogen atoms at the β—and α—anomeric proton of galactose units [18,21]. The chemical shift signals caused by the specific sulfate positions of sulfated polysaccharides from seaweeds are expressed at 3.67–5.31 ppm [17,21,22]. Our results, which were obtained using FTIR and ^1^H NMR analyses, were consistent with those of previous reports on the sulfation of polysaccharides from *C. gigas* [23] and mushroom *Ganoderma atrum* [38]. Furthermore, FTIR and ^1^H NMR results clearly demonstrated that S-SG contains a high amount of sulfate esters when compared with native one. In general, the biological activity of sulfated polysaccharide depends on several structural parameters such as sulfate content [26].

## 4. Materials and Methods

### 4.1. Materials and Chemicals

Red seaweed *G. fisheri* was harvested in October 2020 from the Shrimp Genetic Improvement Center, Surat Thani, Thailand. Human kidney cell line (HK-2) was purchased from ATCC (Manassas, VA, USA). Dulbecco’s modified eagle medium (DMEM) and fetal bovine serum (FBS) were purchased from Gibco, Life Technologies (Grand Island, NY, USA). 3-(4,5-dimethylthiazol-2-yl)-2,5-diphenyltetrazolium bromide (MTT) solution was purchased from PanReac AppliChem, Molecular Probes Inc. (Darmstadt, Germany). 1,1-Diphenyl-2-picrylhydrazyl (DPPH), dihydronicotine amide adenine dinucleotide (NADH), phenazine methosulfate (PMS), nitro blue tetrazolium (NBT), vitamin C, DEAE-sepharose fast flow, dextran standards (DS), phenylmethylsulfonyl fluoride (PMSF), protease inhibitor cocktail, sodium metaperiodate, sodium borohydride, sulphur trioxide-triethylamine complex, *N,N*-dimethylformamide, and deuterium oxide (D_2_O) were purchased from Sigma-Aldrich (St. Louis, MO, USA). A Clarity™ Western ECL Substrate was purchased from the Bio-Rad Laboratories, Inc. (Hercules, CA, USA). Anti-caspase-3, anti-BCL-2, anti-β-actin, and HRP-conjugated goat anti-mouse IgG antibodies were purchased from the Cell Signaling Technology, Inc. (Danvers, MA, USA). Cystone was purchased from the Himaraya drug company (Bangalore, India). All other chemicals were purchased from Merck, Thailand.

### 4.2. Preparation of Sulfated Galactans (N-SG) and Their Derivatives (L-SG and S-SG)

#### 4.2.1. Native Sulfated Galactans (N-SG) from *G. fisheri*

The extraction and purification of N-SG from *G. fisheri* were performed as described by Wongprasert et al. [17]. Briefly, dry seaweed was crushed and mixed with benzene and acetone. *G. fisheri* powder was then dissolved in distilled water and stirred at 35–40 °C for 4 h. This mixture was diluted with hot water and centrifuged at 6000× *g* for 5 min. The supernatant was collected and subsequently kept at −10 °C overnight. After that, the supernatant was centrifuged at 6000× *g* for 5 min. The non-gel fraction was collected and precipitated. The precipitated N-SG was dissolved and centrifuged at 10,000× *g* for 10 min. The supernatant was placed on the DEAE-sepharose fast flow column. The eluted fraction of N-SG was dialyzed before being freeze dried.

#### 4.2.2. Lower Molecular Weight SG (L-SG)

L-SG was obtained by HCl degradation. Briefly, N-SG was dissolved in 0.1 M HCl for 6 h at room temperature. Then, the solution was adjusted to a pH of 8 with NaOH and precipitated 95% ethanol. The pellet was collected after centrifugation at 9500× *g* for 20 min and subsequently resuspended and dialyzed against distilled water in a dialysis bag (MW cutoff 100–500) for 24 h. The L-SG was obtained after freeze drying.

#### 4.2.3. Sulfation SG (S-SG)

S-SG was produced using the periodate oxidation/borohydride reduction method. Briefly, N-SG (6.7 mg/mL) dissolved in distilled water was mixed with sodium metaperiodate (14.3 mg) for 24 h in the dark. After that, ethylene glycol (16.7 mL) was added, and the mixture was stirred for another 2 h. The solution was then subjected to dialysis and freeze-dried to obtain product 1. Product 1 (3.7 mg) was mixed with sodium borohydride (7.4 mg) and dissolved in distilled water for 3 h at room temperature. The solution was then adjusted to a pH of 5 with acetic acid and then to 8 with NaOH. Product 2 was recovered after dialysis and freeze-drying. Product 2 (3.4 mg) was then mixed with a sulphur trioxide–triethylamine complex (16.7 mg) in *N,N*-dimethylformamide (10 mL) for 3 h at room temperature. After that, cold water (12 mL) was added, and the pH was adjusted to 8 and dialyzed against distilled water over 24 h before freeze drying. The resulting product was S-SG.

### 4.3. Evaluation of Antioxidant Activity

#### 4.3.1. DPPH Radical Scavenging Activity Assay

The antioxidant activity of the N-SG/L-SG/S-SG was measured using the DPPH-radical scavenging assay. Various concentrations (0–1000 μg/mL) of the compound solution were mixed with DPPH (200 µM) in methanol. The mixture solution was shaken vigorously, kept in the dark at room temperature for 30 min, and then determined the absorbance at OD 517 nm. Vitamin C was used as an antioxidant positive control. DHHP scavenging activity was expressed as vitamin C equivalent.

#### 4.3.2. Superoxide Radical Scavenging Activity Assay (O_2_^−^)

O_2_^−^ scavenging activity was determined based on the reduction in NBT. The non-enzymatic phenazine methosulfate-nicotinamide adenine dinucleotide (PMS/NADH) system generates free radical superoxide, which reduce NBT to a purple formazan. The 0.5 mL reaction mixture contained phosphate buffer (20 mM, pH 7.4), NADH (73 μM), NBT (50 μM), PMS (15 μM), and various concentrations (0–1000 μg/mL) of N-SG/L-SG/S-SG. After incubation at room temperature for 5 min, the absorbance at OD 560 nm was determined. Vitamin C was used as an antioxidant positive control. The O_2_^−^ radical scavenging capacity was expressed as vitamin C equivalent.

### 4.4. Investigation of Calcium Oxalate Crystallization

Calcium oxalate crystallization was inhibited using a previously described method [50] with slight modifications.

#### 4.4.1. Nucleation Assay

A solution consisting of 50 mmol/l of calcium chloride and 10 mmol/l of sodium oxalate was prepared in a buffer containing 50 mmol/l of Tris–HCl and 150 mmol/l of sodium chloride at pH 6.5. The inhibition activity of the compounds was investigated by mixing 500 µL of calcium chloride solution and 1 mL of the N-SG/L-SG/S-SG at different concentrations (0–1000 µg/mL). Then, crystallization was performed by adding 500 µL of sodium oxalate solution. The temperature was maintained at 37 °C, and the absorbance of the solution was observed at 620 nm every 3 sec for 30 min using a kinetic method. The rate of nucleation was estimated by comparing the induction time in the presence and absence of the compounds. Cystone was used as a positive control. All reactions were performed in triplicate.

The growth of crystals was expected due to the following reaction.
CaCl_2_ + Na_2_C_2_O_4_ → CaC_2_O_4_ + 2NaCl

The percentage inhibition of nucleation was estimated using the following formula.
[1 − (slope sample/slope control)] × 100

#### 4.4.2. Aggregation Assay

Calcium oxalate crystals were prepared. The calcium chloride and sodium oxalate at 50 mmol/l were mixed, equilibrated at 60 °C for 1 h, and then cooled to 37 °C overnight. It was then centrifuged and evaporated to obtain white powder crystals. Calcium oxalate crystals were used at a final concentration of 0.8 mg/mL with a buffer containing 50 mmol/l of Tris–HCl and 150 mmol/l of sodium chloride at pH 6.5. A temperature of 37 °C was maintained throughout the experiment in the presence of the N-SG/L-SG/S-SG at different concentrations (0–1000 µg/mL) and then the solution was further incubated for 30 min. Cystone was used as a positive control. All reactions were performed in triplicate.

The inhibition of nucleation (percentage) was estimated using the following formula.
[(turbidity control-turbidity sample)/turbidity control] × 100

### 4.5. Crystal Morphology

The solution used in the nucleation assay was collected, and 15 µL was dropped onto glass slides. The morphology and arrangement of the calcium oxalate crystals were observed under light microscope (LM) (Leica DFC 7000 T, Leica Microsystem, GmbH, Wetzlar, Germany). The crystal morphology was analyzed in three randomly selected fields at 20× magnification. The crystal with the widest diameter was measured and analyzed using ImageJ. Another 15 µL of the solution was dropped onto the scanning electron microscope (SEM) stubs. After the stubs had dried completely, they were coated with gold for 2 min with a 20 mA coating current. The ultrastructure of the calcium oxalate crystals in the presence of N-SG/L-SG/S-SG was visualized with an SEM (JSM-IT200, JEOL Ltd., Tokyo, Japan).

### 4.6. Protective Effect of S-SG against HK-2 Cell Damage Induced by NaOX

S-SG was chosen to investigate potential protective effects, as it had the highest antioxidants and exhibited the greatest calcium oxalate crystallization inhibition.

#### 4.6.1. Cytotoxicity Detection of Sodium Oxalate (NaOX) and S-SG on HK-2 Cells

HK-2 cells were plated at a density of 1 × 10^4^ cells/well in a 96-well plate in DMEM supplemented with 10% FBS and kept at 37 °C for 24 h. Afterward, the cells were exposed to various concentrations of NaOX (0, 0.156, 0.312, 0.625, 1.25, 2.5, and 5.0 mmol/l) and S-SG (0, 1, 10, 100, 1000, 1500, and 2000 µg/mL) overnight. They were then incubated with an MTT solution at a final concentration of 0.5 mg/mL in each well at 37 °C for 3 h. Subsequently, absorbance was measured at 540 nm.

#### 4.6.2. Effect of S-SG on NaOX-Damaged HK-2 Cells

HK-2 cells in a 96-well plate at density 1 × 10^4^ cells/well were cultured and divided into 6 groups: (1) control—no treatment; (2) NaOX control—treated with 1.25 mmol/l NaOX; (3) 1-S-SG+NaOX—treated with 1 µg/mL S-SG mixed with 1.25 mmol/l NaOX; (4) 10-S-SG+NaOX—treated with 10 µg/mL S-SG mixed with 1.25 mmol/l NaOX; (5) 100-S-SG+NaOX—treated with 100 µg/mL S-SG mixed with 1.25 mmol/l NaOX; (6) 1000-S-SG+NaOX—treated with 1000 µg/mL S-SG mixed with 1.25 mmol/l NaOX. All were allowed to grow overnight. Before cell viability detection, all cell groups were visualized under light microscope for morphological observation. Then, they were incubated with MTT solution at a final concentration of 0.5 mg/mL in each well at 37 °C for 3 h. Subsequently, absorbance was measured at 540 nm.

### 4.7. Determination of Apoptotosis-Related Protein Expression by Western Blot Analysis

Cell damage and death caused by oxalate crystallization-induced apoptosis were evaluated. HK-2 cells were cultured in a 6-well plate at a density 1.6 × 10^5^ cells/well and divided into five groups: (1) control—no treatment; (2) NaOX control—treated with 1.25 mmol/l NaOX; (3) 100-S-SG+NaOX—treated with 100 µg/mL S-SG mixed with 1.25 mmol/l NaOX; (4) 1000-S-SG+NaOX—treated with 1000 µg/mL S-SG mixed with 1.25 mmol/l NaOX; (5) Cystone+NaOX—treated with 100 µg/mL cystone mixed with 1.25 mmol/l NaOX. All were allowed to grow for 24 h and then were harvested. Cells were protein extracted in a protein lysis buffer containing Tris-HCl (20 mM), NaCl (100 mM), PMSF (5 mM), and protease inhibitor coctail (1×). The protein bands were separated on 10–12% SDS-PAGE and transferred to a nitrocellulose membrane. Any nonspecific binding to the membrane was blocked by the addition of 5% nonfat milk in 1× Tris-buffered saline at RT for 2 h and incubated with anti-caspase-3 antibody (1:1000 dilution) and anti-BCL-2 antibody (1:1000 dilution) at 4 °C overnight. Subsequently, the membrane was incubated with a secondary antibody, HRP-conjugated goat anti-mouse IgG (1:2000 dilution). Signals were visualized using a Clarity™ Western ECL Substrate. The positive protein band intensity was measured relative to an internal control (β-actin) using ImageJ (National Institutes of Health, Maryland, MD, USA).

### 4.8. Structural Characterization of S-SG

#### 4.8.1. Sulfate Estimation Analysis

The sulfate content was determined turbidimetrically with barium chloride (BaCl_2_) after HCl hydrolysis. BaCl_2_-gelatin was prepared by dissolving gelatin in 60–70 °C Milli Q water. After 16 h at 4 °C, BaCl_2_ H_2_O was added and dissolved. K_2_SO_4_ was used as a standard sulfate. For determination, the compounds were hydrolyzed for 2 h at 100 °C in 2N HCl. The solutions were then mixed with Milli Q water and 0.5N HCl. Then, BaCl_2_-gelatin was added, swirled, and retained for 30 min at room temperature. Absorbance was subsequently measured at 550 nm. The sulfate percentage was then calculated.

#### 4.8.2. Average Molecular Weight Analysis

The molecular weight was determined using gel permeation chromatography (GPC). The compounds were dissolved in deionization (DI) water and then the solution was performed on a Shimadzu LC system (LC-20A column and RID-10A detector) and a TSKgel guard PWH size exclusion column. DI water was used in the mobile phase at a flow rate of 0.5 mL/min. The column temperature was kept at 60.0 ± 0.1 °C. For MW estimation, the columns were calibrated using dextran standards of known molecular weights (DS-5000, DS-12,000, DS-50,000, DS-80,000, DS-150,000, and DS-270,000). The data were processed using the Class VP program provided by Shimadzu.

#### 4.8.3. Fourier Transform Infrared (FTIR) Spectroscopy Analysis

The compounds were made a transparent film, and the functional groups were recorded by a Bruker TENSOR 27-FTIR spectroscopy in attenuated total reflectance modes with Opus 7.0 software. The spectra were measured in the range of 400–4000 cm^−1^ using a resolution of 1 cm^−1^ over 16 scans.

#### 4.8.4. Nuclear Magnetic Resonance (NMR) Spectroscopy Analysis

The compounds were dissolved in D_2_O in NMR tubes (5 mm in diameter). ^1^H-NMR spectra were recorded by a Varian 400 MHz NMR spectrometer at 80 °C, and ^1^H-NMR chemical shrifts were observed as parts per million (ppm). The spectra were reported relative to an internal D_2_O standard of 4.79 ppm.

### 4.9. Statistical Analysis

All data were expressed as mean ± SEM from three independent experiments. The significance of differences was analyzed using one-way analysis of variance (ANOVA) followed by Turkey’s multiple comparison test on GraphPad Prism version 9. A *p*-value < 0.05 was considered statistically significant.

## 5. Conclusions

We modified sulfated galactans from the red seaweed *G. fisherii* to enhance their sulfate content. The resulting S-SG exhibited strong free radical scavenging activity and the highest antiurolithesis effects of any of the substances tested. It inhibited the nucleation and aggregation of calcium oxalate crystals and modified their morphology. In addition, S-SG was not harmful to HK-2 cells and protected cells from sodium oxalate insult. Furthermore, S-SG also decreased the expression levels of apoptotic caspase-3 and increased anti-apoptotic BCL-2 proteins in HK-2 cells with oxalate-induced damage. The proposed protection of S-SG on HK-2 cell damage induced by sodium oxalate is shown in Figure 8. We propose that S-SG interacts with the oxalate crystal surface and reduces cell–oxalate crystal interaction, thereby protecting cell damage caused by crystals. In addition, S-SG mediates increased BCL-2 upregulation against cell damage. The mechanisms by which S-SG decreased oxalate-induced HK-2 cell apoptosis requires further investigation. Taken together, our findings highlight the potential application of S-SG in the functional food and pharmaceutical industries.

## Figures and Tables

**Figure 1 marinedrugs-20-00382-f001:**
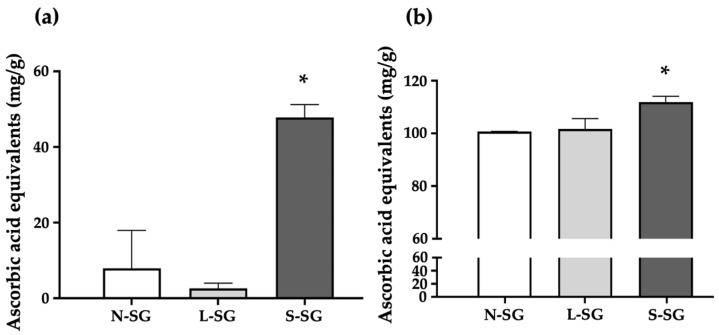
The free-radical scavenging activities of *G. fisheri* N-SG and its derivatives shown as equivalent to ascorbic acid. (**a**) DPPH and (**b**) O_2_^−^. * indicates values that are significantly different from N-SG (*p* < 0.05).

**Figure 2 marinedrugs-20-00382-f002:**
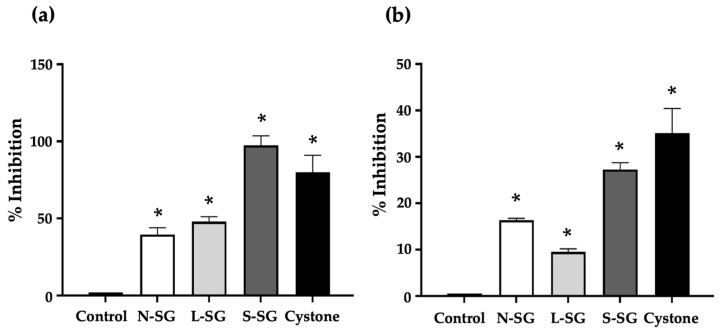
The effects of *G. fisheri* N-SG and its derivatives on (**a**) nucleation inhibition and (**b**) aggregation inhibition of calcium oxalate crystal formation. * indicates values significantly different from control (*p* < 0.05).

**Figure 3 marinedrugs-20-00382-f003:**
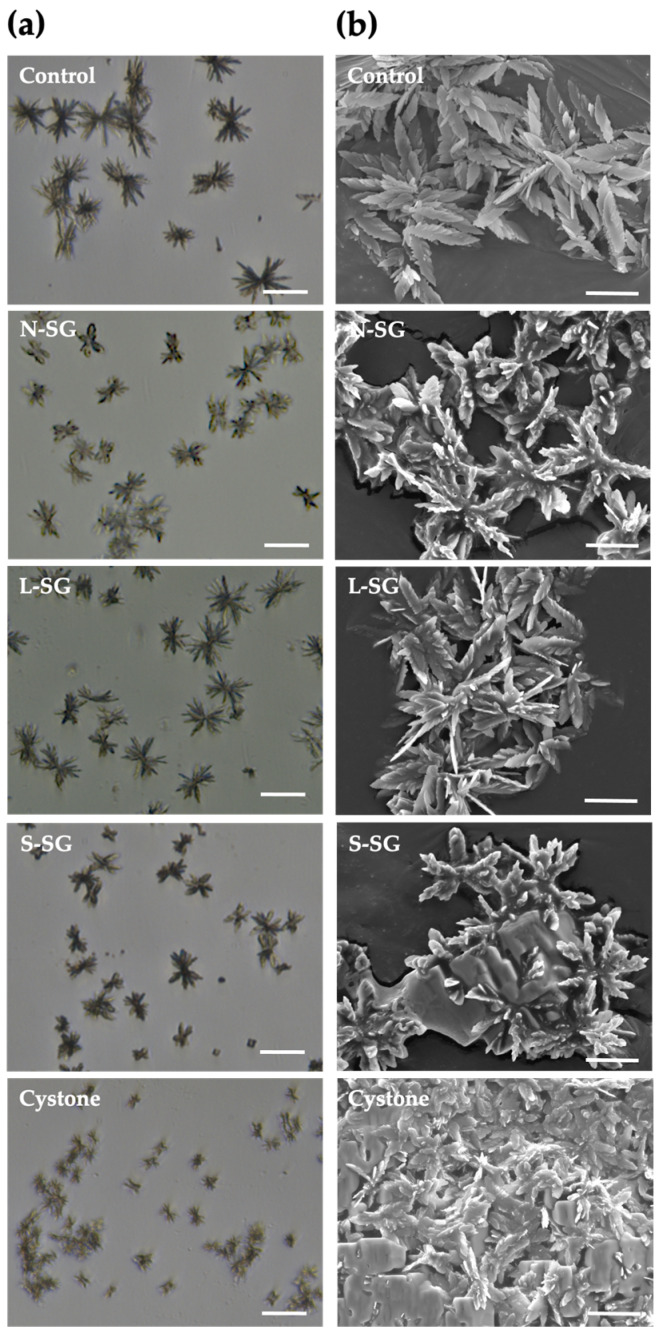
Crystal characterization by (**a**) light microscope (LM) and (**b**) scanning electron microscope (SEM). LM bar = 60 µm; SEM bar = 30 µm.

**Figure 4 marinedrugs-20-00382-f004:**
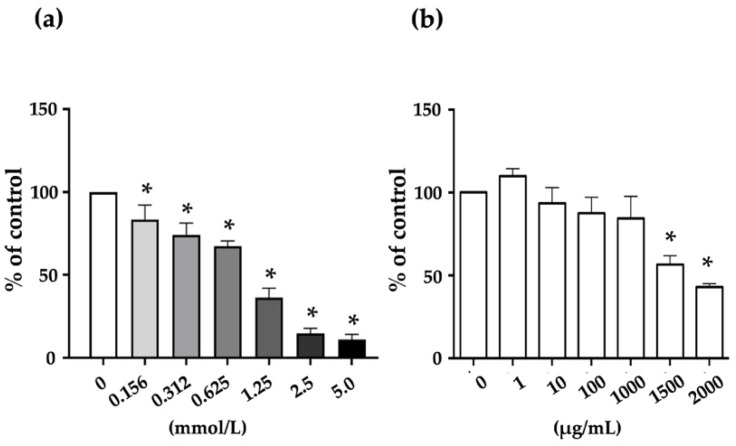
Cytotoxic effects of various concentrations of NaOX (**a**) and S-SG (**b**) on HK-2 cells. * indicates values significantly different from the concentration 0 (*p* < 0.05).

**Figure 5 marinedrugs-20-00382-f005:**
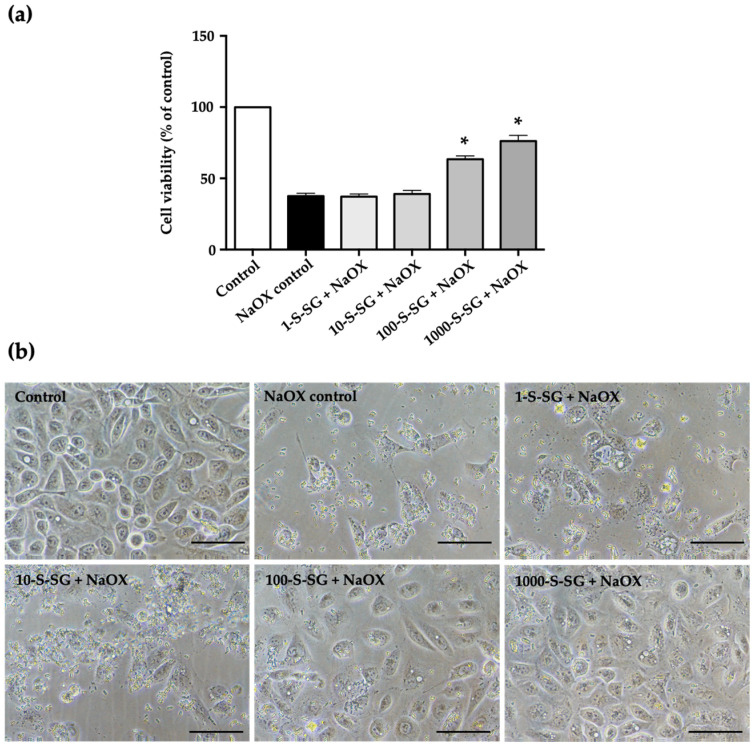
The effect of S-SG on HK-2 cells. (**a**) The protective effect of S-SG against HK-2 cell damage induced by NaOX. (**b**) The morphology of HK-2 cells treated with a mixture of S-SG and NaOX. Bar = 400 um; * indicates values significantly different from control (*p* < 0.05).

**Figure 6 marinedrugs-20-00382-f006:**
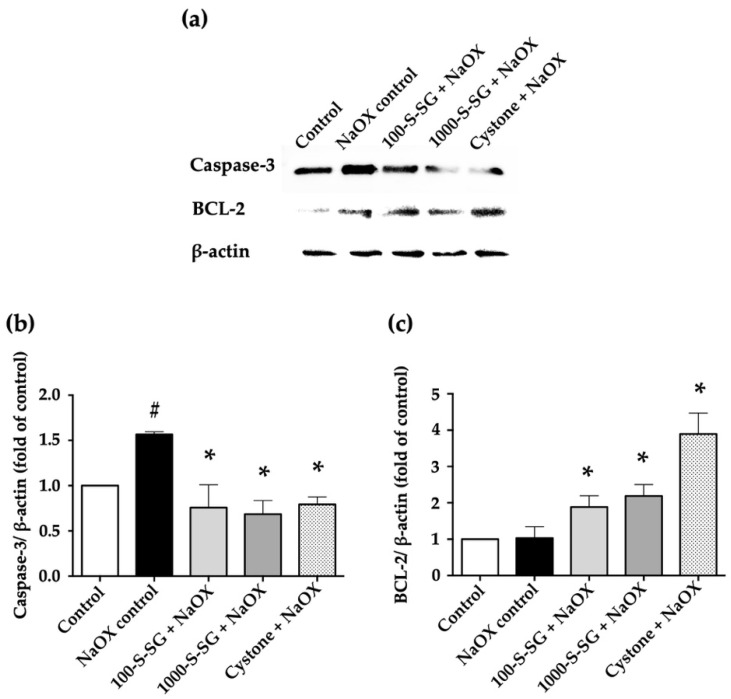
(**a**) Expression levels of apoptotic caspase-3 and anti-apoptotic BCL-2 proteins in HK-2 cells after treatment with S-SG and NaOX for 24 h by Western blot analysis. (**b**) Caspase-3 was normalized to β-actin, and the fold changes over the control were plotted. (**c**) BCL-2 was normalized to β-actin, and the fold changes over the control were plotted. # indicates values that are significantly different from control, and * indicates values that are significantly different from NaOX control (*p* < 0.05).

**Figure 7 marinedrugs-20-00382-f007:**
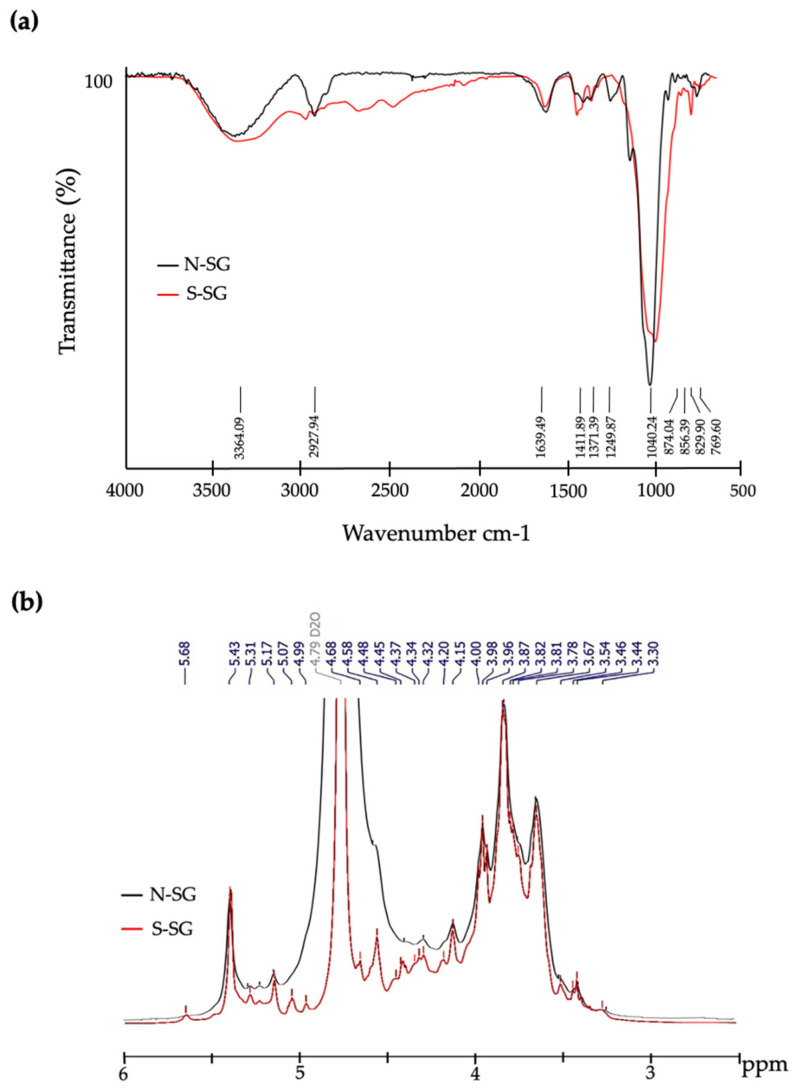
Comparative structures of N–SG and S–SG from *G. fisheri*. (**a**) FTIR spectra and (**b**) ^1^H NMR spectra.

**Figure 8 marinedrugs-20-00382-f008:**
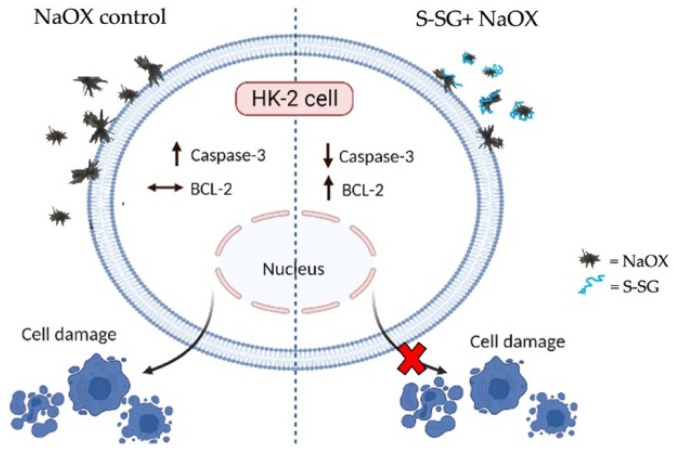
A schematic diagram showing the proposed protection of S-SG from sodium oxalate-induced HK-2 cell damage. S-SG, which is due to high sulfate groups, possibly interacts with oxalate crystal surface, reduces cell–oxalate crystal interaction, and subsequently decreases HK-2 cell damage. Furthermore, S-SG also decreases expression of caspase-3 and increases expression of BCL-2, resulting in reduced cell damage induced by sodium oxalate crystals (created with BioReder.com).

**Table 1 marinedrugs-20-00382-t001:** The effects of *G. fisheri* N-SG and its derivatives on calcium oxalate crystal diameters and the percentage presence of calcium oxalate monohydrate (COM) and calcium oxalate dihydrate (COD) crystallizations. * indicates values significantly different from the concentration 0 (*p* < 0.05).

	Diameter of Crystals (µm)	Crystallizations (%)	
		COM 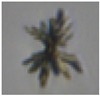	COD 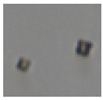
Control	84.35 ± 3.16	99.10 ± 0.85	0.90 ± 0.84
N-SG	68.58 ± 6.70 *	96.77 ± 1.46	3.23 ± 1.46
L-SG	79.93 ± 3.02 *	95.00 ± 2.38 *	5.00 ± 1.25 *
S-SG	65.59 ± 6.71 *	95.50 ± 0.21 *	4.50 ± 0.22 *
Cystone	26.50 ± 5.00 *	87.92 ± 3.46 *	12.08 ± 3.29 *

## Data Availability

Not applicable.

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
