# Peer review of "Increased Sulfation in Gracilaria fisheri Sulfated Galactans Enhances Antioxidant and Antiurolithiatic Activities and Protects HK-2 Cell Death Induced by Sodium Oxalate"

_marinedrugs, 2022, doi:10.3390/md20060382_

Round 1

Reviewer 1 Report

The study investigates the in vitro antioxidant and antiurolithiatic activities of native sulfated galactans of the red seaweed Gracilaria fisheri as well as modified ones by increased sulfatation. The study brings new data on the potential antiurolithiatic activity of modified sulfated galactans giving evidence of their capacity to inhibit the nucleation and aggregation of calcium oxalate crystals as well as their morphology and size. Experimental section is well designed and results are clearly presented.

Few recommendations

  • When first mentioned in the text the Genus should be given completely and not only the first letter and add the type of organism (seaweed, mussel, fungi) for the given examples in the introduction and discussion
  • Add statistics to table 1
  • In Materials and methods: the information on origin of the seaweed should be given (site and period of collection, samples from cultivation or wild environment). Also, add a Solvent and Reagent section to give information about suppliers.

Author Response

Thank you for your suggestion, please find the attachment that we have responded to your comments point by point

Reviewer 2 Report

In this study, natural Gracilaria fisheri sulfated galactans (N-SG) with molecular weight of 217.4 kDa and its modified derivatives were investigated. With such a large molecular weight, it was difficult to enter cells. How did they protect HK-2 cells from death induced by sodium oxalate?

In addition, the structure of polysaccharides needs to be confirmed.

Not suitable for publication.

Author Response

(The authors gave the same response as above.)

Reviewer 3 Report

Gracilaria fisheri is a red algae naturally present along the coastal areas of Thailand. Literature data report that the galactans sulfates (SG) isolated from this red alga exhibit immunostimulatory and antiviral activities. In this work, native galactans sulfates from G. fisheri were modified and subjected to further sulfation. Subsequently, the in vitro antioxidant and antiurolytic activities of the modified substances and their ability to protect against sodium oxalate-induced renal tubular cell death (HK-2) were investigated.

The topic approaches a little investigated aspect of the beneficial role of the increase in sulfation of galactans sulphates for the antiurolytic activity.

The experimental part supports the design of the work by FTIR spectroscopy and NMR analysis and is validated by statistical analysis.

The references reflect the state of international knowledge, are sufficiently recent and provided in adequate quantities.

On the basis of the results obtained, correct conclusions were drawn.

I have no requests for changes to this manuscript.

Author Response

Thank you for your critical reviewing and accepting our manuscript.

Reviewer 4 Report

Thank you for the invitation to review this manuscript.

This study, of   Waraporn Sakaew et al.  describes the in vitro antioxidant and antiurolithiatic activities of the modified galactans from Gracilaria fisheri and their ability  to protect against sodium oxalate-induced renal tubular (HK-2) cell death.

I have found the manuscript interesting and comprehensive with a lot of data. The authors are specific to the relevant objectives of the study. All the assays are correctly described and informed.

The results and discussion section is well managed and the experimental data well described.

The manuscript may increase the knowledge in the field of the antioxidants and antiurolithiatic of natural origin.

Author Response

(The authors gave the same response as above.)

Round 2

Reviewer 2 Report

The author basically replied to my concerns and recommended publication.